# On the Construction and Evaluation of Color Invariant Networks/ Conference Submissions

## Abstract

This is an empirical paper which constructs color invariant networks and evaluates their performances on a realistic data set. The paper studies the simplest possible case of color invariance: *invariance under pixel-wise permutation of the color channels*. Thus the network is aware not of the specific color object, but its *colorfulness*. The data set introduced in the paper consists of images showing crashed cars from which ten classes were extracted. An additional annotation was done which labeled whether the car shown was red or non-red. The networks were evaluated by their performance on the classification task. With the color annotation we altered the color ratios in the training data and analyzed the generalization capabilities of the networks on the unaltered test data. We further split the test data in red and non-red cars and did a similar evaluation. It is shown in the paper that an pixel-wise ordering of the *rgb*-values of the images performs better or at least similarly for small deviations from the true color ratios. The limits of these networks are also discussed.

## 1 Introduction

Imagine a training set without red objects, and a test set which contains red objects. How well does a trained net perform? This is not a mere academic question. Imagine we want to separate cars, humans and free space for an autonomous driving task. If our data contained red cars but not red trousers say, it will most likely classify legs as cars. Even worse it could mix up yellow markings and yellow trouser and classify an human as free space. On the other hand we can not disregard color all together as it yields some clues for natural objects such as trees, sky, mist, snow and also some man made objects such as markings, or traffic signs. The first thing that comes to mind is to balance the color statistics of our data set. But this impossible to do in practice, and worse at training time it is unknown which colors will become fashion in say five years. What is called for is network which is invariant under color changes. In this paper we construct and analyze such a network.

### 1.1 Structure of the paper

The paper begins with some remarks on the literature in Section 2. This is followed by Section 3 which discusses different variants of color invariant networks and evaluates them on *cifar10*. Section 4 is the main contribution of the paper. The *crashed cars* data set is introduced. And and the best color invariant function of the previous section is evaluated on this data set. The paper closes with a conclusion in Section 5. In Appendix A more details on the data sets can be found. This is followed by Appendix B which collects some plots and figures which did not fit in the main text. Finally, Appendix explores the limits of color invariance.

## 2 Notes on the literature

In this review section we first discuss invariance in general, then we point to some recent papers which discuss various variants of invariance. This is followed by a brief discussion of equivariance. Then we point the reader to color invariance. Finally we mention some work on the evaluation of invariance.

Let us denote our data by $X$, our target space by $Y$ and our network by $\phi \colon X \to Y$. In the paper $X$ is an image and $Y$ the finite set of classes. To discuss *invariance*, we consider a transformation $T \colon X \to X$ on our data space. We say that a net is *invariant* under such a transformation $T$ provided that $\psi(x) = \psi(T(x))$.

Putting invariance in a network is an classical question of neural networks. In Section 8.7 of Bishop (1997) we find the following advises to ensure invariance: *by example*, *by pre-processing*, *through structure*. If the training data has some invariance it is hoped that the net learns to this invariance. We can enforce this by cropping at random, flipping or adding noise to the training images. Color normalization falls into preprocessing, the images are invariant under (some) changes of intensity or luminosity. Structural invariance could for example be enforced by radial basis functions, which may not apply directly to images. A different structural invariance is max pooling. Through a cascade of max pooling layers the networks become invariant under small movements of the image plane. Together with cropping this ensures heuristically the translation invariance in image classification. In Bishop (1997) we find some further hints to the classical literature.

Another popular recent architecture which is related to invariance are spatial transformer networks introduced by Jaderberg et al. (2015). Here the transformer network is specialized to normalize the input image by scaling, translating, rotating and so on. This makes the input data empirically stable under such transformations. More general, all types of normalizing layers can be seen as such an invariance enforcing unit. In Shen et al. (2017) translation and rotation invariance nets are constructed by patch reordering based on some energy. What is nice about this approach is that this extends the local spatial invariance of max-pooling to a much larger scale. Let us also mention Li (2017) in which rotational invariant networks are constructed.

Invariance is closely related to *nuisances*. Nuisances are properties of the object which are irrelevant to classification. They are in depth discussed in Soatto (2011). Typically they are countered by extending the data set as recommended above. This is done for example in Dosovitskiy et al. (2017) with computer models of chairs or in Pasquale et al. (2016) with real images. Their exists many more examples.

Sometimes it is more desirable that the net is aware of these transformation. This can be achieved by the related concept of equivariance. Contrary to invariance, equivariant networks are aware of the transformation, thus there exists a transformation $T'$ on the target space such that $\psi(T(x)) = T'\psi(x)$. Ideally, both transformations are a groups, such as one of the wallpaper groups. It is possible to extend convolution and pooling to the group setting. For more on this we refer to Cohen & Welling (2016).

There exists a deep theory of color invariance derived from physical principles, see for example Geusebroek et al. (2001). An performance evaluation of color invariance was done by Burghouts & Geusebroek (2009). We cite these papers to remind us that color invariance is much more just the invariance under pixel permutation of the color channels as discussed in the present paper. Typically, classical papers discussing color invariants look at invariance under color changes of the SIFT features.

A more recent empirical study of invariance of deep neural nets can be found in Goodfellow et al. (2009). In the paper the idea is promoted that invariance in deep neural nets can only be empirically evaluated by activation of neurons. This evaluation scheme is applied for example by Shen et al. (2017). Finally, Peng et al. (2014) followed a different path by evaluating invariance with synthetic images.

## 3  COMPARISON OF INVARIANT FUNCTIONS

### 3.1  PIXEL-WISE COLOR INVARIANT FUNCTIONS

In this paper we say that a function is *pixel-wise color invariant* if a permutation of the color channels of any pixel does not change the outcome of the function. For what follows we will simply speak of a *color invariant* function, when we actually mean pixel-wise color invariant. It thus suffices to discuss invariance of a function which depends on three parameters. There are several ways to make such a function invariant under permutation of its inputs. Formally we can write this as $p(x, y, z) = p(\sigma(x, y, z))$. In this paper we analyzed the symmetric polynomials $p_1(x, y, z) =$

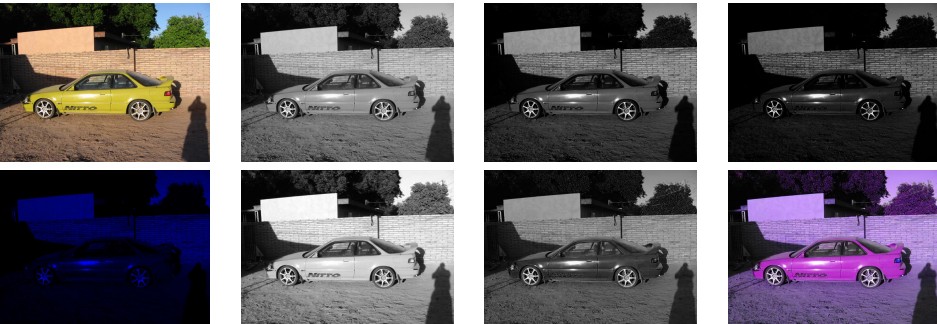

Figure 1: The invariant functions discussed in paper applied to the yellow car on the upper left. In the first row from the left: original image, the first symmetric polynomial, the second symmetric polynomial, the third symmetric polynomial. In the second row: all symmetric polynomials, pixel-wise maximum, pixel-wise minimum and the ordered network

$x + y + z$, $p_2(x,y,z) = xy + yz + yz$ and $p_3(x,y,z) = xyz$, and variants of sorting: $q_1 = \max\{x,y,z\}, q_2 = \min\{x,y,z\}, q_3 = \text{sort}\{x,y,z\}$. It is obvious that these functions are invariant under permutations of its inputs. So for instance $p_1(x,y,z) = p_1(y,z,x)$ and so on. We could also consider linear combinations of these function. We did this only for the symmetric functions. In Figure 1 we applied all permutation invariant functions to the yellow car shown on the upper left.

## 3.2 EVALUATION ON CIFAR-10

*Cifar10* introduced by Krizhevsky (2009) is a popular data set which has been chosen for the analysis in this section. As *baseline* we implemented *tensorflow's cifar10* architecture, as can be found at Google. So the baseline net, let us denote it by $\psi$, takes an image $x$ and outputs its class $y = \psi(x, w)$. The invariant nets are almost similar. Instead of passing the image directly to the net, the images are made invariant by first applying one of the functions mention above followed by the architecture of the baseline. So, $y = \psi(p(x), \tilde{w})$. We call the different architectures the $p_1$-*net*, the $p_2$-*net* and so on. Figure shows a sketch of a similar net architecture. The network architecture is similar to the tutorial and can be obtained there. No parameter other than the stopping time has been changed.

**Training.** We trained the net for 249999 iterations. This is longer as suggested in the tutorial, but as we did not change any of the training parameters we gave the nets some more time to converge. Table shows the accuracy after 249999 iterations.

Table 1: The table shows the accuracy after 249999 iterations on the validation set. We see that the ordering network performs comparable to the standard *cifar10* implementation. All other architecture fail.

| iteration | cifar10 | sym | $p_1$ | $p_2$ | $p_3$ | min | max | ord |
|---|---|---|---|---|---|---|---|---|
| 249999 | 0.865 | 0.836 | 0.839 | 0.778 | 0.766 | 0.833 | 0.836 | 0.864 |

**Results and Discussion.** Comparing the gray net (the $p_1$-net) to the baseline we see a drop in accuracy. We interpret this that color gives some clues to the classifier and can not entirely be omitted. Still it is not a dominant factor as the drop is not that dramatic. The other *symmetric* functions have a significant drop in accuracy. The accuracy did not increase if all *three symmetric* functions were considered. Surprisingly, the *maximum* and the *minimum* function contain lots of information. Finally, *sorting* achieved similar results on the test set as the baseline. So it seems that colorfulness of the input image provides enough information for classification.

Table 2: This lists the ten most frequent body type classes of the crash data set.

| class | body type |
|-------|-----------|
| 00 | 5-door/4-door hatchback |
| 01 | 2-door sedan, hardtop, coupe |
| 02 | 3-door/2-door hatchback |
| 03 | Large utility |
| 04 | Large pickup |
| 05 | Station Wagon |
| 06 | Compact pickup |
| 07 | Large utility |
| 08 | Minivan |
| 09 | 4-door sedan, hardtop |

# 4 EVALUATION ON THE CRASHED CARS DATA SET

The previous section showed that a net which pixel-wise orders the color channels performs similar to a baseline net on *cifar10*. Let us call such a net an *order network*. In a second set of more involved experiments we compared *order networks* to a *baseline* on a more realistic data set. To this end we extracted a data set from the *NHTSA* which compiles accident reports during the years 2004-2010 in the US, United States Department of Transportation (2017). The data set was split in train and test set and consists of around 158000 images of 37000 cars. These car are categorized by the *NHTSA* in several body type classes. The task of the nets was to classify the body type of the car from the image.

In addition we annotated whether the car shown in the image is red or not. With this additional annotation we could fix the ratio of red cars in the training data and perform some experiments with varying color analyzing the invariance properties of the nets. More on the data set is described in the appendix.

## 4.1 THE CRASHED CARS DATA SET

*Since 1972, NCSA's Special Crash Investigations (SCI) Program has provided NHTSA with the most in- depth and detailed level of crash investigation data collected by the agency.*[1] For this paper we obtained images from the website of the agency. These images show crashed cars from the years 2004 - 2010. Each image is provided with the body type of the car shown, Table 2. We selected full view images of the ten most frequent body types. Figure 2 shows one image of each body type class.

## 4.2 NETWORK ARCHITECTURES

As the baseline model we have chosen an *alexnet* type network Krizhevsky et al. (2012). We did some experiments, not reported here, which varied the size of the net, the size of the fully connected layer, and the points of weight regularization. As a result of these experiments we reduced the size of the fully connected layers to 256 instead of usual 4096, added batch normalization, and $l_2$-regularization on the weights of the last layer to reduce over-fitting. The reason for choosing *alexnet* was due to its reasonable convergence time which allowed several experiments.

**Network architectures.** Figure 3 shows the principal architectures of the networks analyzed in the paper. The baseline network, a variant of *alexnet*, inputs an image of a car and outputs the corresponding class. The color invariant networks of the paper, are of identical structure, except of an additional *inv-block*. The *inv block* is applied to the image and then passed through the *alexnet* architecture outputting the corresponding class. In this paper two variants have been analyzed, one which just orders the *rgb*-values of each pixel, and a second which applies first an $3x3x3x3$ convolution to the input image, and then orders pixel-wise.

---

[1]From the website of the special crash investigations, United States Department of Transportation (2017)

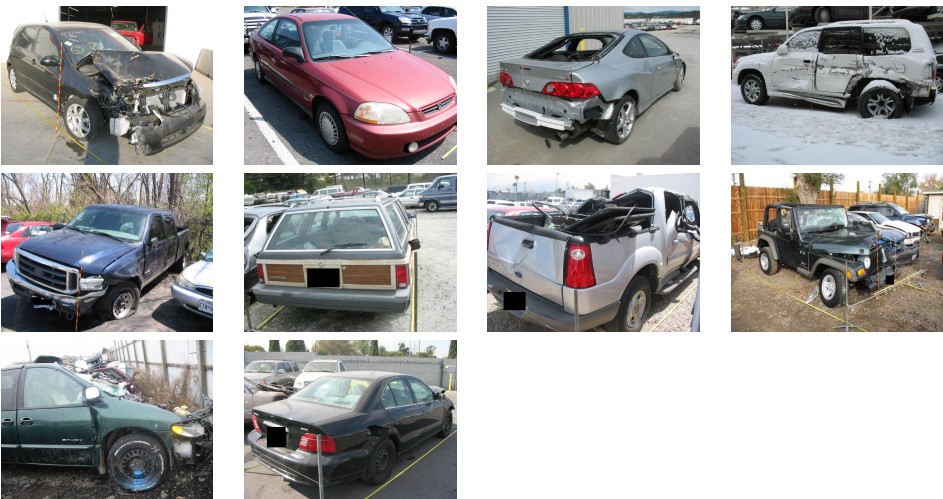

Figure 2: The figure show some images of the NHTSA data set, one for each class.

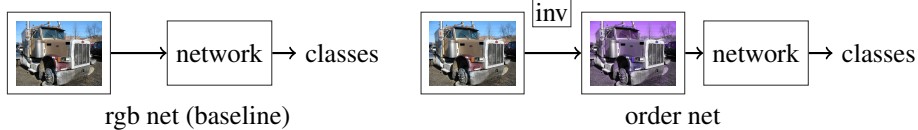

rgb net (baseline)                                     order net

Figure 3: A sketch of the baseline architecture shown on the left and the order architectures shown on the right. On the right the images is first passed through an inv-block an then to a network identical to the baseline network. The inv-block either orders the *rgb*-values of each pixel, or applies a $3x3x3x3$ convolution to the image and then orders each pixel. The network block is in all cases an identical variant of *alexnet*.

**Naming conventions.**   In this paper we denote the standard nets by *rgb-nets*. We call nets which pixel-wise order the *rgb* channels *order nets*. Finally, the nets which apply a color correction to the image before ordering are called *weighted order nets*.

### 4.3 TRAINING AND TESTING

**Training sets.**   As described in the appendix, we created three groups of training sets, one which contained all cars denoted by *all-train*, one which contained no red cars, denoted by *nored-train* and one with uniform ratio of red / non-red cars per class, denoted by *even-train*. All three nets were trained on the mentioned data sets for 25000 iterations. We did not optimize to train for the specific data, but are interested in the effects if the color distribution changes. In an uncontrolled setting, we are not aware of the specific color distribution, and we can only decide to stop training with the data at hand - it is this what we are mimicking here.

**Testing sets.**   In total we created four groups of test sets. One denoted by *all-test* consisting of a sample of all cars. A second group denoted by *nored-test* containing no red cars. The third group called *red-test*, consisting of one hundred sets of red cars only. And the fourth group, called *class-test*, splitting color and class giving thirty further tests sets. As the distribution of red cars is not uniform over the classes we sub-sampled the first three sets such that each class has the same number of cars. This is not entirely satisfying as rarer classes contained more views of the same car, than larger classes. But this is the best we could do.

### 4.4 EXPERIMENTS ON THE BASELINE

For the baseline we trained the nets on *all-train* and evaluated them on our four test sets.

**Tested on all-test.** Figure 4 shows the plots of accuracy over iteration. We see from the figure the reason for choosing to stop at 25000 as at this point the rgb-net performs best beating its competitors. But if we look below at the class experiments we see that all three nets perform similarly. As all numbers are in similar range we see that our sub-sampling estimated the true accuracy rather well.

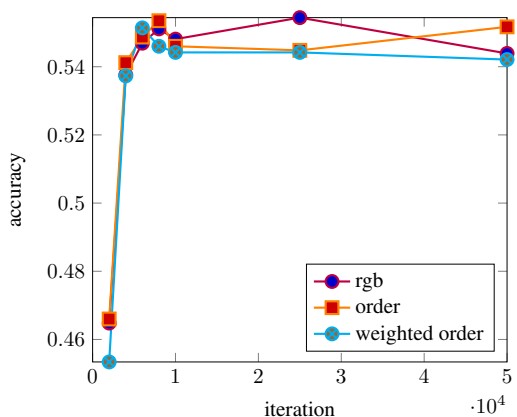

| iteration | rgb | order | worder |
|-----------|--------|--------|--------|
| 2000 | 0.4648 | 0.4660 | 0.4534 |
| 4000 | 0.5376 | 0.5412 | 0.5373 |
| 6000 | 0.5469 | 0.5487 | 0.5514 |
| 8000 | 0.5511 | 0.5535 | 0.5460 |
| 10000 | 0.5481 | 0.5460 | 0.5442 |
| 25000 | 0.5544 | 0.5448 | 0.5442 |
| 50000 | 0.5439 | 0.5517 | 0.5421 |

Figure 4: The net was trained on *all-train* and tested on *all-test*. This shows the accuracy over the iteration in table and plot. The highlighted row shows the 25000 iteration at which all results are analyzed.

**Tested on nored-test** Figure 11, in the appendix, shows the plots of accuracy over iteration. Similar to the previous paragraph we see that all three nets behave similar, by again taking the class experiments into account.

**Tested on red-test** The achieved accuracy of the trained nets, were tested on 100 sampled sets consisting of red cars only. We see that the order nets perform 0.03 absolute units better than the baseline net on red cars. Figure 5 shows three histograms of the accuracies. The accuracy of the *rgb network* is $0.5113 \pm 0.0306$, the accuracy of the *order network* is $0.5420 \pm 0.0284$, and an accuracy of *weighted order network* of $0.5411 \pm 0.0280$.

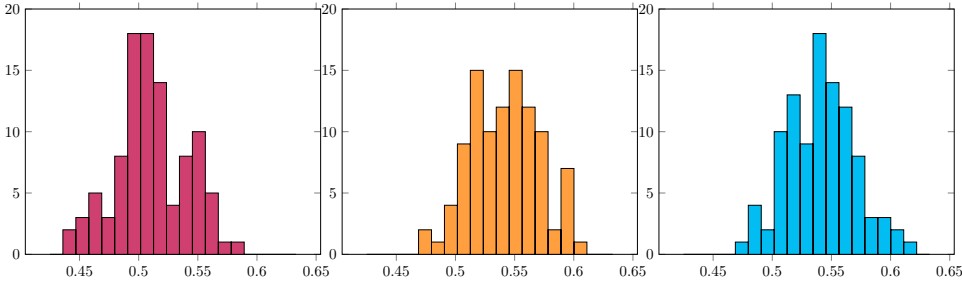

Figure 5: The nets were train on *all-train* and evaluated on *red-test*. The figure on the shows the histogram of one hundred sampled subsets of the test data containing only red cars. This results in an accuracy of *rgb network* of $0.5113 \pm 0.0306$, an accuracy of the *order network* of $0.5420 \pm 0.0284$, and an accuracy of *worder network* of $0.5411 \pm 0.0280$.

**Tested on class-test** Figure 6 shows the heat plot of the accuracies per class and test set. From these numbers we may derive the mean and deviation and compare them to the accuracies computed in the previous paragraph. As can be seen the numbers are in a similar range, confirming the conclusions of the previous three experiments.

For the readers convenience we derived all the means and standard deviations. The *rgb net* on all cars has accuracy $0.551 \pm 0.105$. The *order net* on all cars has accuracy $0.553 \pm 0.103$. The *weighted*

*order net* on all cars has accuracy $0.550 \pm 0.115$. The *rgb net* on all non-red cars has accuracy $0.551 \pm 0.106$. The *order net* on all non-red cars has accuracy $0.554 \pm 0.101$. The *weighted order net* on all non-red cars has accuracy $0.551 \pm 0.111$. The *rgb net* on all red cars has accuracy $0.513 \pm 0.123$. The *order net* on all red cars has accuracy $0.545 \pm 0.099$. The *weighted order net* on all red cars has accuracy $0.538 \pm 0.137$.

| all cars | | | non-red cars | | | red-cars | | |
|---|---|---|---|---|---|---|---|---|
| 0.432 | 0.37 | 0.362 | 0.401 | 0.398 | 0.383 | 0.414 | 0.336 | 0.289 |
| 0.487 | 0.459 | 0.43 | 0.469 | 0.45 | 0.412 | 0.559 | 0.512 | 0.494 |
| 0.449 | 0.465 | 0.416 | 0.448 | 0.456 | 0.43 | 0.43 | 0.504 | 0.355 |
| 0.556 | 0.602 | 0.59 | 0.555 | 0.598 | 0.591 | 0.484 | 0.547 | 0.57 |
| 0.622 | 0.666 | 0.647 | 0.626 | 0.675 | 0.642 | 0.599 | 0.643 | 0.656 |
| 0.436 | 0.502 | 0.529 | 0.469 | 0.482 | 0.518 | 0.289 | 0.609 | 0.539 |
| 0.702 | 0.664 | 0.66 | 0.692 | 0.65 | 0.658 | 0.703 | 0.68 | 0.68 |
| 0.591 | 0.586 | 0.568 | 0.595 | 0.588 | 0.572 | 0.568 | 0.584 | 0.543 |
| 0.74 | 0.704 | 0.749 | 0.744 | 0.713 | 0.747 | 0.672 | 0.609 | 0.766 |
| 0.499 | 0.516 | 0.552 | 0.512 | 0.529 | 0.561 | 0.413 | 0.423 | 0.489 |

Figure 6: This shows the accuracies as heatmap of the thirty test classes of the nets trained on *all-train*. Each row represents on of the bodytype classes numbered 00 to 09. The columns of the blocks represent the *rgb-network* the *order network* and the *weighted order* network. The block on the left are the accuracies computed on all cars. The block in the middle the non-red cars, and the block on the right are the accuracies computed on all red cars.

## 4.5 EXPERIMENTS ON NON-RED IMAGES

For our next analysis we trained the nets on a data set without red cars, denoted *nored-train* in the paper. All plots except for the heat map can be found in the appendix B.2. We see from the plots that the weighted order net beats both other nets on all cars and on all non-red cars by 0.01 to 0.02 absolute units. On the red cars the order nets beats the other architectures significantly with 0.08 absolute units. The achieved accuracy of the trained nets, were tested on 100 sampled sets consisting of red cars only. Figure 14 shows three histograms of the accuracies. The accuracy of the *rgb network* is $0.3707 \pm 0.0272$, the accuracy of the *order network* is $0.4583 \pm 0.0281$, and an accuracy of *weighted order network* of $0.3864 \pm 0.0280$. A plot of the heat map is shown in Figure 7.

| all cars | | | non-red cars | | | red-cars | | |
|---|---|---|---|---|---|---|---|---|
| 0.307 | 0.388 | 0.346 | 0.336 | 0.346 | 0.409 | 0.234 | 0.422 | 0.297 |
| 0.516 | 0.465 | 0.512 | 0.515 | 0.483 | 0.508 | 0.52 | 0.379 | 0.545 |
| 0.286 | 0.343 | 0.367 | 0.325 | 0.333 | 0.431 | 0.172 | 0.367 | 0.199 |
| 0.6 | 0.633 | 0.572 | 0.637 | 0.655 | 0.617 | 0.297 | 0.477 | 0.211 |
| 0.59 | 0.626 | 0.591 | 0.623 | 0.657 | 0.644 | 0.432 | 0.469 | 0.359 |
| 0.516 | 0.504 | 0.502 | 0.526 | 0.505 | 0.484 | 0.336 | 0.531 | 0.453 |
| 0.614 | 0.593 | 0.675 | 0.64 | 0.61 | 0.684 | 0.484 | 0.539 | 0.645 |
| 0.554 | 0.496 | 0.569 | 0.58 | 0.506 | 0.595 | 0.354 | 0.422 | 0.396 |
| 0.692 | 0.71 | 0.7 | 0.709 | 0.726 | 0.733 | 0.438 | 0.5 | 0.391 |
| 0.533 | 0.524 | 0.512 | 0.544 | 0.537 | 0.533 | 0.459 | 0.441 | 0.37 |

Figure 7: This shows the heatmap of the thirty test classes of the nets trained on *nonred-train*. As a further analysis we may count the times the net beat its competitors. This shows again that the *order networks* perform better than the *baseline*.

## 4.6 EXPERIMENTS ON UNIFORMLY DISTRIBUTED COLOR RATIOS

Annotating the images with red / non-red allowed several further experiments. Here we fixed the ratio of red to non-red cars per class and report the achieved accuracies. Figure 8 shows the plots of our analysis. For comparison also the baseline as trained and evaluated in Section 4.4 was included in the plots. We see that up to a ratio of 0.4 all nets behave acceptable, with the *order nets* beating

the *rgb net*. After that the accuracies of all nets drop rapidly. As we are randomly choosing from the test data set, this can also be due to the fact that these cars show only a fraction of the whole dataset.

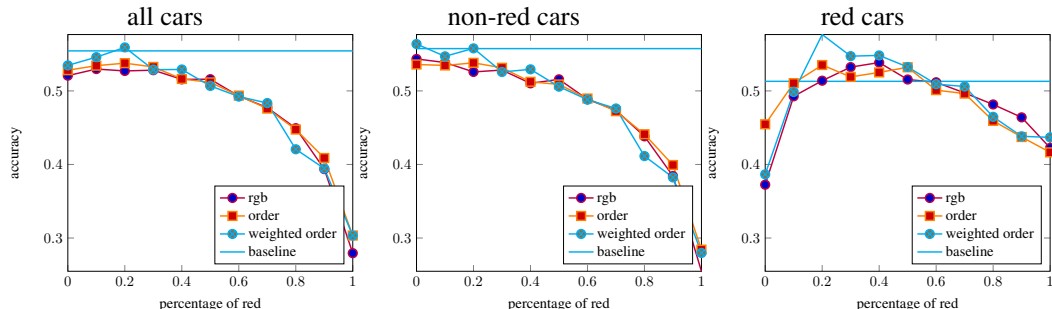

Figure 8: The net were trained on eleven training sets each with a fixed ratio of red / non-red cars per class. In the paper we called this data set *even-train*. Starting with no red car at $0$ on the $x$-axis and only red cars at $1$ on the $x$-axis. The nets were than evaluated on *all-test*, *non-test* and *red-test*. The baseline was trained on *all-train* and then evaluated on the corresponding data sets.

To analyze this behavior further we computed the deviation of ratios of red cars in *even-train* to the true ratio as shown in Table 4. In Figure 9 we see that at 0.2 the color ratio in *even-train* is closest to the true ratio. Looking again at Figure 8 we see that at this ratio the weighted nets excel on all data sets, this is due to the color adjustment in the weights. The *order net* also beats the baseline at this point. Furthermore we see that if we are not to far away from the true ratio, that is between 0.0 and 0.4 the accuracies of the order nets are similar or better than that of the *rgb nets*.

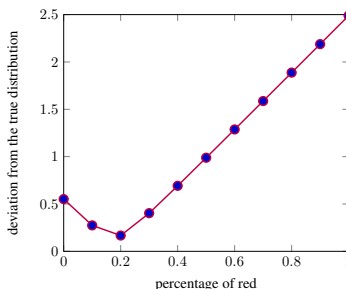

Figure 9: The plot show the deviation from the true color distribution per class while changing the ratio of red to non-red car.

## 4.7    AT THE LIMITS

In Appendix C we also report our experiments which trained on a set without red cars except for class 09 which contained red cars only. In the plots shown we see the expected behavior - all nets are not able to learn non-red cars in class 09 and have trouble of detecting red cars in the classes 00-08. We are to far away from the true color ratios, with a deviation of $0.99$, and thus the nets fail.

## 5    SUMMARY AND CONCLUSIONS

We compared different color invariant neural networks. It is shown in the paper that only pixel-wise ordering of the color channels shows similar results on *cifar10* (and also on the crashed car data set). To test the hypothesis that ordering is invariant under color changes, a classification task has been extracted from a publicly available crashed car data set. In addition each car was labeled as red or non-red. On this data set it was shown that all three nets showed similar behavior on all cars and on all non-red cars, on the red cars the order nets performed noticeably better. Further, we excluded

red cars from the training set, and showed that the weighted order nets performed better than the baseline on all three test sets. On the red cars the order showed significantly better results. Further, we fixed the ratio of red / non-red in the training sets. The order nets perform better or at least similar to the baseline net. All nets degrade noticeable while increasing the ratio red cars. As a teaser we report in the appendix there all three nets fail. No net can cope with one class of entirely red cars and all other classes set to non red.

We can also view the paper as an empirical study on generalization: trained nets are tested on a statistically different test set. Most plots of accuracy over iterations on the test set showed overshooting despite of the $l_2$ regularization in the final layer. The curve of the weighted net in Figure 4 being a typical example. We interpret this as over fitting, training should be stopped much earlier. A further empirical conclusion shows that sub-sampling the unevenly distributed test data gave similar results than deriving the accuracies for all class separately and then taking the mean. But the individual class may perform rather poor, an insight which is lost in sub-sampling.

The paper introduced and evaluated a variant of color invariant nets. The constructed nets are invariant under pixel-wise permutation of the color channels. Thus the network is aware not of the specific color, but the *colorfulness* of the object. Further, a data set was introduced which allowed to evaluate color invariance in a realistic setting. We see that the net constructed in the paper are better or equal to the baseline if the color distribution is not to far away from the true distribution. We conclude that colorfulness is enough information for classification. The crash car data set itself calls for further experiments and insights, and remains a tough classification challenge.

## 6 LITERATURE

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

# A    APENDIX - THE CRASHED CAR DATA SET

In this appendix we report some additional information on the crashed car data set. In particular, we briefly discuss why classifying crashed cars is a hard task, give some details on the selection of the cars, the labeling, and on the construction of the training and testing sets.

**Why it is a hard task.**    The classification task considered in the paper is hard, as it is sometimes just not possible to decide to which class a shown image belongs. Further complications arise from the data set itself, as it shows sometimes also completely destroyed images, or close up and so on. We did not omitted such images from the data set. Figure 10 shows some of the particular challenges the net has to overcome. We did not strive to excel at this task, therefore we did not investigate this any further.

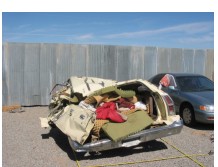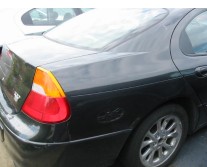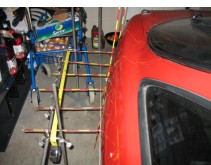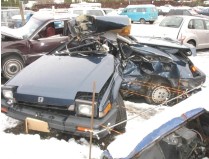

Figure 10: The figure shows two images of crashed cars and two close-ups.

**Details on car selection.**    Each case comes along with an *xml* file giving some meta information. Since we are not interested in say close-ups of the cars, we picked the images with the annotations: *Frontleftoblique, Frontrightoblique, Backleftoblique, Backrightoblique*, showing the views of the whole car. We further included the labels: *side, left, right, overhead, front, back, down, middoor, oblique* and excluded the labels *tire, mirror, handle, filler, wheel, door, visor, dash, seat, tank, grass, tin, gauge*.

## A.1    LABELING RED CARS

For the paper we manually annotated whether the car shown is red and non-red.

**Details of labeling.**    We looked at several views of a car and manually labeled whether the car is red or not. We did this for the training data set as well as for the test data. In addition we experimented by selecting three pixel of the image showing the color of the car, but we decided against it for this paper, as this color detection should again be checked against a manually labeled data set.

**Label error and noise.**    We estimate our label error to be 0.01. The estimation was done by re-evaluating the labeled non red images and counting the number of red images found. So in reality one out of 100 cars in the images is actually red. Furthermore, potentially there are red cars in the background or red cars appears through reflection.

## A.2    TRAINING DATA SETS

For training we took images from the years 2004-5, 2007-10. In total there are 31918 cars and 135843 images in the training set. We generated three different groups of training sets. The first group denoted by *all-train* consists of an evenly distributed sample of the training data of all cars. The second group denoted by *nonred-train* consists of an evenly distributed sample of the training data of all non-red cars. The third group denoted by *even-train* consists of randomly chosen images from the training data with a fixed frequency of non-red to red images per class.

Some images were not blacklisted due to errors in the *jpg*. In the statistics below such images are still included.

**The set all-train.**    For each class we sampled 2500 images from the training data. This lead in total to 25000 images collected in the set *all-train*. Sneaking at the statistics we see that there are 2861 images of class 00 in our training data. This lead to our present choice of samples size.

**The set nored-train.** The data set *nored-train* consists of a sample of 2404 non red images per class.

**The set even-train.** In total there are ten data sets denoted by *even010*, ... , *even100*. For each class we collected all non-red images and all red images. To generate the data sets, we sampled $p * N$ from the red images and $(1 - p) * N$ from the non red images. In our experiments we have set $N = 3000$ and varied $p$ from 0.1 to 1.0.

**Overall statistics of training data.** In Table 3 we list the statistics of the training data. Table 4

Table 3: The table list the number of cars per class in the training set, the percentage of the cars, as well as the number and percentages of red cars per class.

| class | 00 | 01 | 02 | 03 | 04 | 05 | 06 | 07 | 08 | 09 | total |
|---|---|---|---|---|---|---|---|---|---|---|---|
| all | 670 | 2584 | 901 | 1226 | 2676 | 827 | 1601 | 5037 | 1779 | 14617 | 31918 |
| % | 0.02 | 0.08 | 0.03 | 0.04 | 0.08 | 0.03 | 0.05 | 0.16 | 0.06 | 0.46 | 1. |
| red | 121 | 566 | 214 | 154 | 494 | 98 | 302 | 648 | 173 | 1589 | 4359 |
| % | 0.18 | 0.22 | 0.24 | 0.13 | 0.18 | 0.12 | 0.19 | 0.13 | 0.10 | 0.11 | 0.14 |

shows the number of images per class of the training data. We observe that all ratios are in a similar range. Each car has roughly four different views, 14 % of all cars and images are red.

Table 4: The table list the number of images per class in the training set, the percentage of the cars, as well as the number and percentages of red cars per class.

| class | 00 | 01 | 02 | 03 | 04 | 05 | 06 | 07 | 08 | 09 | total |
|---|---|---|---|---|---|---|---|---|---|---|---|
| all | 2861 | 10898 | 3919 | 5197 | 11342 | 3597 | 6940 | 21492 | 7582 | 62015 | 135843 |
| % | 0.2 | 0.08 | 0.03 | 0.04 | 0.08 | 0.03 | 0.05 | 0.16 | 0.06 | 0.46 | 1. |
| red | 516 | 2392 | 981 | 673 | 2163 | 453 | 1256 | 2828 | 731 | 6679 | 18762 |
| % | 0.18 | 0.22 | 0.25 | 0.13 | 0.19 | 0.13 | 0.18 | 0.13 | 0.10 | 0.11 | 0.14 |

### A.3 TEST DATA SETS

Only images from the year 2006 were used in the creating of the test data. In total, there are 5381 cars from the year 2006 leading to 22794 images. Each car has on average four different views. In total we created four different groups of test sets. The first group denoted by *all-test* consists of an evenly distributed sample of all classes. Similarly, the second group denoted by *nored-test* consists of an evenly distributed sample of all non-red classes. The third group denoted by *red-test* consists of one hundred evenly distributed sampled images of red classes. The fourth group encoded by *allNN-test*, *redNN-test*, *nonredNN-test* are similar to the three previous groups, except that all images of a specific class are taken. So for example *red09-test* consists of all images of red cars of class 09. The next paragraphs explain the four groups in more detail. A further paragraph lists the frequencies of cars per class.

**The set all-test.** The set *all-test* consists of a sample of 323 images per class of all cars of the test set. In total there are 3230 images in this set. As a motivation for these number we look at the statistics of the test set and see that there are 323 images in class 00.

**The set nonred-test.** The set *nonred-test* consists of a sample of 273 images per class of non-red cars of the test set. In total there are 2730 images in this set. The limiting number is again the number of non-red images of class 00.

**The sets red-test.** Looking again at the statistics we see that there are 25 red images in class 05. This would result in a rather small test set. Therefore we sampled one hundred sets consisting of 25 images per class. In total each of the sets consists of 250 images.

**The sets class-test.** The fourth group are several partitions of the test data set. The images *all00-test ... all09-test* combined, are all images of the test sets. Likewise *nonred00-test ... nonred09-test* are all non-red images of the test set. Finally, *red00-test ... red09-test* are all red images of the test set. Looking at the statistics we see for example that *red03-test* consists of 78 images and so on.

**Overall statistics of test data.** Table 5 lists the number of cars per classes, the percentages with respect to all cars, as well as the number of red cars of the test set. We see that most cars in the class 09 the *4-door sedan*. Furthermore, we read of the table that sport cars, which typically have three doors and thus fall in class 02, are more likely to be red than the average car.

Table 5: The table list the number of cars per class in the test set, the percentage of the cars, as well as the number of red cars per class and ration of red cars per class.

| class | 00 | 01 | 02 | 03 | 04 | 05 | 06 | 07 | 08 | 09 | total |
|---|---|---|---|---|---|---|---|---|---|---|---|
| all | 75 | 514 | 177 | 190 | 485 | 93 | 287 | 854 | 302 | 2404 | 5381 |
| % | 0.01 | 0.10 | 0.03 | 0.04 | 0.09 | 0.02 | 0.05 | 0.16 | 0.06 | 0.45 | 1. |
| red | 12 | 100 | 42 | 18 | 86 | 7 | 55 | 110 | 28 | 293 | 751 |
| % | 0.16 | 0.19 | 0.24 | 0.09 | 0.18 | 0.08 | 0.19 | 0.13 | 0.09 | 0.12 | 0.14 |

Table 6 lists the number of images per class. We see that the percentages of cars and image coincides.

Table 6: The table list the number of images per class in the test set, as well as the number and percentages of images of red cars.

| class | 00 | 01 | 02 | 03 | 04 | 05 | 06 | 07 | 08 | 09 | total |
|---|---|---|---|---|---|---|---|---|---|---|---|
| all | 323 | 2206 | 799 | 759 | 2013 | 388 | 1221 | 3707 | 1284 | 10094 | 22794 |
| % | 0.01 | 0.10 | 0.04 | 0.03 | 0.09 | 0.02 | 0.05 | 0.16 | 0.06 | 0.44 | 1. |
| red | 50 | 409 | 184 | 78 | 363 | 25 | 249 | 481 | 118 | 1232 | 3189 |
| % | 0.15 | 0.18 | 0.23 | 0.10 | 0.18 | 0.06 | 0.20 | 0.13 | 0.09 | 0.12 | 0.14 |

# B   FURTHER PLOTS

## B.1   FURTHER PLOTS OF ALL-TRAIN

In this appendix we show further plots which did not fit in the paper. Figure 11 shows the plots of the nets trained on *all-train* and tested on *nonred-test* of Section 4.4.

| iteration | rgb | order | worder |
|---|---|---|---|
| 2000 | 0.4663 | 0.4684 | 0.4478 |
| 4000 | 0.5302 | 0.5465 | 0.5249 |
| 6000 | 0.5579 | 0.5501 | 0.5586 |
| 8000 | 0.5565 | 0.5572 | 0.5558 |
| 10000 | 0.5320 | 0.5600 | 0.5568 |
| 25000 | 0.5575 | 0.5497 | 0.5511 |
| 50000 | 0.5575 | 0.5504 | 0.5476 |

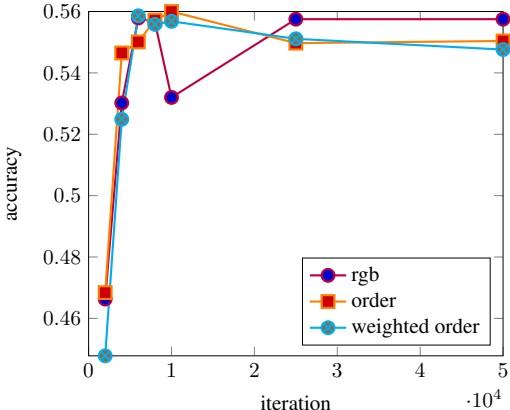

Figure 11: The net was trained on *all-train* and tested on *nored-test*. This shows the accuracy over the iteration in table and plot. The net was trained on all cars and tested on the non red ars. The highlighted row shows the 25000 iteration at which all results are analyzed.

## B.2 FURTHER PLOTS OF NORED-TRAIN

The remaining plots of Section are shown in Figure 12, Figure 13, and Figure 14.

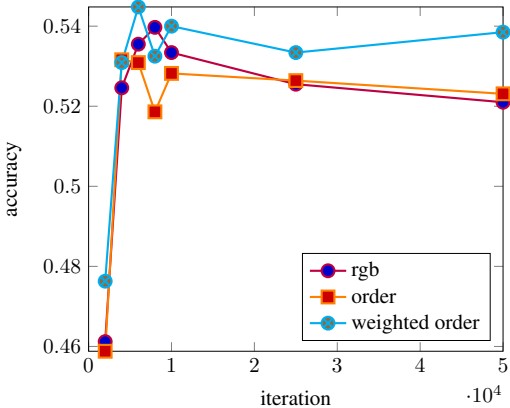

| iteration | rgb | order | worder |
|---|---|---|---|
| 2000 | 0.4612 | 0.4588 | 0.4763 |
| 4000 | 0.5246 | 0.5316 | 0.5309 |
| 6000 | 0.5355 | 0.5309 | 0.5448 |
| 8000 | 0.5397 | 0.5186 | 0.5325 |
| 10000 | 0.5334 | 0.5282 | 0.5400 |
| 25000 | 0.5255 | 0.5264 | 0.5334 |
| 50000 | 0.5210 | 0.5231 | 0.5385 |

Figure 12: The net was trained on *nored-train* and tested on *all-test*. This shows the accuracy over the iteration in table and plot. The highlighted row shows the 25000 iteration at which all results are analyzed.

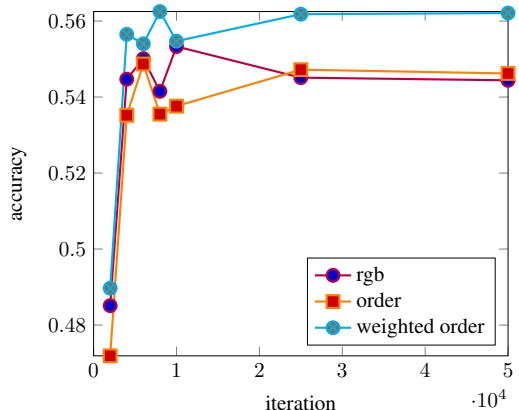

| iteration | rgb | order | worder |
|---|---|---|---|
| 2000 | 0.4851 | 0.4719 | 0.4897 |
| 4000 | 0.5447 | 0.5352 | 0.5565 |
| 6000 | 0.5501 | 0.5487 | 0.5540 |
| 8000 | 0.5415 | 0.5355 | 0.5625 |
| 10000 | 0.5533 | 0.5376 | 0.5547 |
| 25000 | 0.5451 | 0.5472 | 0.5618 |
| 50000 | 0.5444 | 0.5462 | 0.5621 |

Figure 13: The net was trained on *nored-train* and tested on *nored-test*. This shows the accuracy over the iteration in table and plot. The net was trained on all cars and tested on the non red ars. The highlighted row shows the 25000 iteration at which all results are analyzed.

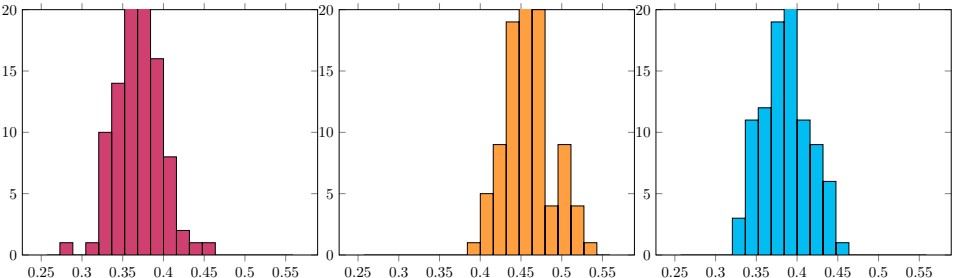

Figure 14: The nets were train on *nored-train* and evaluated on *red-test*. The figure on the shows the histogram of one hundred sampled subsets of the test data containing only red cars. This results in an accuracy of *rgb network* of $0.3707 \pm 0.0272$, an accuracy of the *order network* of $0.4583 \pm 0.0281$, and an accuracy of *worder network* of $0.3864 \pm 0.0280$.

## C    AT THE LIMITS

### C.1    REPORT ON ALLRED-IN09-TRAIN

As a teaser we want to report also a case at which non of the nets perform that well if we take a closer look. In this setting there are no red cars in the training set except for class 09 which consists of red cars only. A first look at the performance of all cars we see a overall drop in accuracy by roughly 0.10 absolute units, Figure 15. Tested on non-red cars the performance was better, with a drop by 0.05 absolute units, Figure 16. Looking at the color histograms already reveals that all nets behave poorly on the red cars, Figure 17. Finally, what is really happening is visible in Figure 18. The red cars of class 09 are classified almost with perfection. On the other classes the performance of all nets is poor.

As we are randomly choosing from the test data set, this could also be due to the fact that these cars show only a fraction of the whole dataset.

| iteration | rgb | order | worder |
|---|---|---|---|
| 2000 | 0.3993 | 0.4044 | 0.3663 |
| 4000 | 0.4387 | 0.4450 | 0.4177 |
| 6000 | 0.4483 | 0.4555 | 0.4423 |
| 8000 | 0.4498 | 0.4579 | 0.4345 |
| 10000 | 0.4486 | 0.4597 | 0.4288 |
| 25000 | 0.4483 | 0.4522 | 0.4279 |
| 50000 | 0.4483 | 0.4582 | 0.4351 |

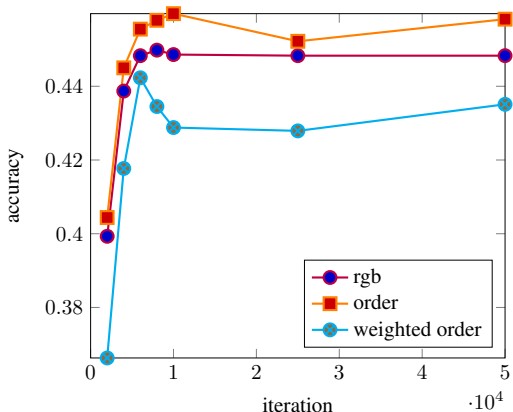

Figure 15: The net was trained on *allred-in09-train* and tested on *all-test*. This shows the accuracy over the iteration in table and plot. The highlighted row shows the 25000 iteration at which all results are analyzed.

| iteration | rgb | order | worder |
|---|---|---|---|
| 2000 | 0.4386 | 0.4418 | 0.4006 |
| 4000 | 0.4822 | 0.4964 | 0.4624 |
| 6000 | 0.4908 | 0.5004 | 0.4741 |
| 8000 | 0.4996 | 0.5057 | 0.4719 |
| 10000 | 0.4826 | 0.5000 | 0.4666 |
| 25000 | 0.4901 | 0.5000 | 0.4709 |
| 50000 | 0.4929 | 0.5028 | 0.4751 |

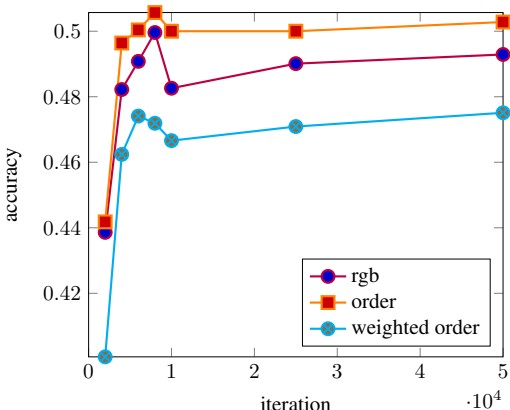

Figure 16: The net was trained on *allred-in09-train* and tested on *nored-test*. This shows the accuracy over the iteration in table and plot. The net was trained on all cars and tested on the non red ars. The highlighted row shows the 25000 iteration at which all results are analyzed.

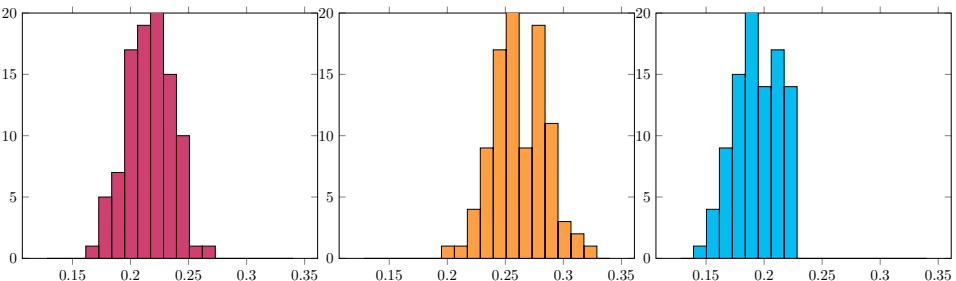

Figure 17: The nets were train on *allred-in09-train* and evaluated on *red-test*. The figure on the shows the histogram of one hundred sampled subsets of the test data containing only red cars. This results in an accuracy of *rgb network* of $0.2166 \pm 0.0198$, an accuracy of the *order network* of $0.2620 \pm 0.0224$, and an accuracy of *worder network* of $0.1940 \pm 0.0181$.

| all cars | | | non-red cars | | | red-cars | | |
|---|---|---|---|---|---|---|---|---|
| 0.318 | 0.31 | 0.266 | 0.391 | 0.346 | 0.323 | 0.023 | 0.062 | 0.039 |
| 0.447 | 0.485 | 0.447 | 0.55 | 0.592 | 0.55 | 0.01 | 0.023 | 0.008 |
| 0.308 | 0.33 | 0.276 | 0.38 | 0.42 | 0.344 | 0.051 | 0.094 | 0.035 |
| 0.527 | 0.458 | 0.53 | 0.566 | 0.499 | 0.572 | 0.203 | 0.18 | 0.125 |
| 0.573 | 0.595 | 0.538 | 0.653 | 0.657 | 0.605 | 0.221 | 0.323 | 0.24 |
| 0.471 | 0.451 | 0.479 | 0.508 | 0.482 | 0.497 | 0.078 | 0.086 | 0.039 |
| 0.524 | 0.61 | 0.47 | 0.614 | 0.68 | 0.551 | 0.211 | 0.332 | 0.172 |
| 0.541 | 0.575 | 0.547 | 0.593 | 0.613 | 0.593 | 0.209 | 0.332 | 0.23 |
| 0.676 | 0.693 | 0.64 | 0.72 | 0.724 | 0.691 | 0.266 | 0.359 | 0.172 |
| 0.123 | 0.134 | 0.123 | 0.017 | 0.035 | 0.018 | 0.884 | 0.843 | 0.878 |

Figure 18: The nets were trained on *allred09-train*. This shows the accuracy as measured on the thirty *class* data sets. Each row represents on of the bodytype classes numbered 00 to 09. The columns of the blocks represent the *rgb-network* the *order network* and the *weighted order* network. The block on the left are the accuracies computed on all cars. The block in the middle the non-red cars, and the block on the right are the accuracies computed on all red cars.

