# OpenReview forum: "On the Construction and Evaluation of Color Invariant Networks"
_ICLR.cc/2018/Conference — Reject_

### Official Review · AnonReviewer1 · 2017-11-26
**Review for "On the Construction and Evaluation of Color Invariant Networks"**

**Rating:** 4
**Confidence:** 4

**Review:**

The paper proposes and evaluates a method to make neural networks for image recognition color invariant.

The contribution of the paper is:
 - some proposed methods to extract a color-invariant representation
 - an experimental evaluation of the methods on the cifar 10 dataset
 - a new dataset "crashed cars"
 - evaluation of the best method from the cifar10 experiments on the new dataset

Pros:
 - the crashed cars dataset is interesting. The authors have definitely found an interesting untapped source of interesting images.


Cons:
- The authors name their method order network but the method they propose is not really parts of the network but simple preprocessing steps to the input of the network.
- The paper is incomplete without the appendices. In fact the paper is referring to specific figures in the appendix in the main text.
 - the authors define color invariance as a being invariant to which specific color an object in an image does have, e.g. whether a car is red or green, but they don't think about color invariance in the broader context - color changes because of lighting, shades, ..... Also, the proposed methods aim to preserve the "colorfullness" of a color. This is also problematic, because while the proposed method works for a car that is green or a car that is red, it will fail for a car that is black (or white) - because in both cases the "colorfulness" is not relevant. Note that this is specifically interesting in the context of the task at hand (cars) and many cars being, white, grey (silver), or black.
- the difference in the results in table 1 could well come from the fact that in all of the invariant methods except for "ord" the input is a WxHx1 matrix, but for "ord" and "cifar" the input is a "WxHx3" matrix. This probably leads to more parameters in the convolutions.
- the results in the  figure 4: it's very unlikely that the differences reported are actually significant. It appears that all methods perform approximately the same - and the authors pick a specific line (25k steps) as the relevant one in which the RGB-input space performs best. The proposed method does not lead to any relevant improvement.
Figure 6/7: are very hard to read. I am still not sure what exactly they are trying to say.

Minor comments:
 - section 1: "called for is network" -> called for is a network
 - section 1.1: And and -> And
 - section 1.1: Appendix -> Appendix C
 - section 2: Their exists many -> There exist many
 - section 2: these transformation -> these transformations
 - section 2: what does "the wallpaper groups" refer to?
 - section 2: are a groups -> are groups
 - section 3.2: reference to a non-existing figure
 - section 3.2/Training: 2499999 iterations = steps?
 - section 3.2/Training: longer as suggested -> longer than suggested

---

### Official Review · AnonReviewer2 · 2017-11-28
**Rejection of badly written paper with very limited experimental proof.**

**Rating:** 3
**Confidence:** 4

**Review:**

The authors investigate a modified input layer that results in color invariant networks. The proposed methods are evaluated on two car datasets. It is shown that certain color invariant "input" layers can improve accuracy for test-images from a different color distribution than the training images.


The proposed assumptions are not well motivated and seem arbitrary. Why is using a permutation of each pixels' color a good idea?

The paper is very hard to read. The message is unclear and the experiments to prove it are of very limited scope, i.e. one small dataset with the only experiment purportedly showing generalization to red cars.

Some examples of specific issues:
- the abstract is almost incomprehensible and it is not clear what the contributions are
- Some references to Figures are missing the figure number, eg. 3.2 first paragraph,
- It is not clear how many input channels the color invariant functions use, eg. p1 does it use only one channel and hence has fewer parameters?
- are the training and testing sets all disjoint (sec 4.3)?
- at random points figures are put in the appendix, even though they are described in the paper and seem to show key results (eg "tested on nored-test")
- Sec 4.6: The explanation for why the accuracy drops for all models is not clear. Is it because the total number of training images drops? If that's the case the whole experimental setup seems flawed.
- Sec 4.6: the authors refer to the "order net" beating the baseline, however, from Fig 8 (right most) it appears as if all models beat the baseline. In the conclusion they say that weighted order net beats the baseline on all three test sets w/o red cars in the training set. Is that Fig 8 @0%? The baseline seems to be best performing on "all cars" and "non-red cars"

In order to be at an appropriate level for any publication the experiments need to be much more general in scope.

---

### Official Review · AnonReviewer3 · 2017-11-30
**The paper lacks with respect to novelty, clarity of presentation, theoretical and practical motivation, as well as results.**

**Rating:** 3
**Confidence:** 4

**Review:**

The authors test a CNN on images with color channels modified (such that the values of the three channels, after modification, are invariant to permutations).

The main positive point is that the performance does not degrade too much. However, there are several important negative points which should prevent this work, as it is, from being published.

1. Why is this type of color channel modification relevant for real life vision? The invariance introduced here does not seem to be related to any real world phenomenon. The nets, in principle, could learn to recognize objects based on shape only, and the shape remains stable when the color channels are changed.

2. Why is the crash car dataset used in this scenario? It is not clear to me why this types of theoretical invariance is tested on such as specific dataset. Is there a real reason for that?

3. The writing could be significantly improved, both at the grammatical level and the level of high level organization and presentation. I think the authors should spend time on better motivating the choice of invariance used, as well as on testing with different (potentially new) architectures, color change cases, and datasets.

4. There is no theoretical novelty and the empirical one seems to be very limited, with less convincing results.

---

### Decision · Program_Chairs · 2018-01-29
**ICLR 2018 Conference Acceptance Decision**

**Decision:**

Reject

**Comment:**

Three reviewers recommend rejection and there is no rebuttal.